# Cells on Hydrogels with Micron-Scaled Stiffness Patterns Demonstrate Local Stiffness Sensing

**DOI:** 10.3390/nano12040648

**Published:** 2022-02-15

**Authors:** Abbas Mgharbel, Camille Migdal, Nicolas Bouchonville, Paul Dupenloup, David Fuard, Eline Lopez-Soler, Caterina Tomba, Marie Courçon, Danielle Gulino-Debrac, Héléne Delanoë-Ayari, Alice Nicolas

**Affiliations:** 1University Grenoble Alps, CNRS, LTM, 38000 Grenoble, France; mgharbelabs@hotmail.com (A.M.); camille.migdal@cellandsoft.com (C.M.); Nicolas.bouchonville@gmail.com (N.B.); pdupenlp@stanford.edu (P.D.); david.fuard@cea.fr (D.F.); lopez.eline@gmail.com (E.L.-S.); caterina.tomba@univ-lyon1.fr (C.T.); 2University Grenoble Alps, CEA, CNRS, Inserm, BIG-BCI, 38000 Grenoble, France; marie.courcon@cea.fr (M.C.); gulino.danielle@gmail.com (D.G.-D.); 3University Grenoble Alps, CNRS, Grenoble INP, Institut Néel, 38000 Grenoble, France; 4Université de Lyon, University Claude Bernard Lyon1, CNRS, Institut Lumière Matière, 69622 Villeurbanne, France; helene.delanoe-ayari@univ-lyon1.fr

**Keywords:** stiffness patterning, gray-leveled lithography, traction force microscopy, hydrogel, cell ADHESION, mechanosensitivity

## Abstract

Cell rigidity sensing—a basic cellular process allowing cells to adapt to mechanical cues—involves cell capabilities exerting force on the extracellular environment. In vivo, cells are exposed to multi-scaled heterogeneities in the mechanical properties of the surroundings. Here, we investigate whether cells are able to sense micron-scaled stiffness textures by measuring the forces they transmit to the extracellular matrix. To this end, we propose an efficient photochemistry of polyacrylamide hydrogels to design micron-scale stiffness patterns with kPa/µm gradients. Additionally, we propose an original protocol for the surface coating of adhesion proteins, which allows tuning the surface density from fully coupled to fully independent of the stiffness pattern. This evidences that cells pull on their surroundings by adjusting the level of stress to the micron-scaled stiffness. This conclusion was achieved through improvements in the traction force microscopy technique, e.g., adapting to substrates with a non-uniform stiffness and achieving a submicron resolution thanks to the implementation of a pyramidal optical flow algorithm. These developments provide tools for enhancing the current understanding of the contribution of stiffness alterations in many pathologies, including cancer.

## 1. Introduction

Cell sensitivity to the mechanical properties of an extracellular environment is presently seen as a promising route for stem cell engineering and regenerative medicine [1,2], as well as the design of organs on a chip [3]. Many technologies have arisen that build soft environments for cell cultures. Their aim is either to mimic the in-vivo mechanical environment [4] or to enhance a specific cellular response, such as cell differentiation using cell mechanosensitivity [5]. To this end, soft hydrogels or elastomers have been developed with specific mechanical properties to support 2D or 3D cell culture. The most common are polyacrylamide hydrogels with optimized protein coatings [6] that reproduce the soft compliance of animal tissues, in the kPa range [7]. Polyacrylamide may be replaced by other hydrogels, such as polyethylene glycol (PEG) derivatives or by soft polydimethylsiloxane (PDMS) for processing purpose. These substrates, dedicated to 2D culture, are at the origins of seminal works on stem cell mechanosensitivity [8,9] and cancer invasiveness [10]. Other techniques focus on the design of soft 3D supports for cell culture, which either process natural hydrogels with pore sizes larger than the cell sizes [11], or, as more recently seen, take the benefit from 3D bioprinting technologies [12]. These developments have brought important advances in the design of organoids, with a high level of physiological mimicry [13], as well as in the understanding of fundamental processes in cell migration, cancer invasion, stem cell differentiation, etc. [14,15,16]. Aside these technological developments, the mechanical characterization of human tissues has progressed down to the micro- and nanoscales (see [17] and references therein), which are the scales the cells first probe in their interactions with the extracellular environment. Cell receptors are a few nanometers in size and they organize into adhesive micron-sized clusters that are involved in the mechanical probing of the extracellular environment [18]. Complex nano- and micro-scale patterns of rigidity, with very sharp kilopascals per micrometers (kPa/µm), and gradients at the subcellular scales, have been evidenced in human tissues [19,20], as it is expected from the composition of an extracellular environment that consists of an entanglement of different molecular components with different mechanical properties [5]. Recently, few technologies have addressed the design of theses sharp gradients and their impact on cells, using appositions of materials with distinct mechanical properties [21,22], photodegradation of chemical bonds [23], electron-beam induced reticulation [24], or a posteriori photoreticulation of regions in a well formed 3D scaffold [25]. From these studies, stiffness patterning appears to influence cell behavior. For instance, stem cell differentiation was shown to be sensitive to the patterned stiffness and to the spatial organization of the stiff patterns [23,24]. Although the effects on the cellular responses are beginning to be reported on, the length scales in which the stiffness signals are integrated are currently not known. It is well documented that cells sense the surrounding rigidities by pulling on their anchorages, with adhesion and the actomyosin cytoskeleton as major players. Signaling at the molecular level, e.g., in focal adhesion, is presently largely documented (see [26] and references therein). However, several studies have also highlighted the role of cell-scaled contractility mediated by molecular motors, which is at the origin of the traction forces that cells apply on the extracellular environment [27,28,29,30,31,32].

Here, we use a mechanical approach to investigate cell capabilities to sense stiffness variations at the micrometer scale. Our goal is to analyze how the tensile forces that the cells exert on the extracellular environment adapt to subcellular stiffness variations. To this end, we set up a photo-polymerization process dedicated to polyacrylamide, which allowed printing subcellular, micron-scaled, modulations of the rigidity with kPa/µm gradients, in the range of compliances that are reported in human tissues. To really assess cell sensitivity to these micron-scaled stiffness heterogeneities, one major concern is to decouple the surface density of the adhesive ligands on the substrate from the stiffness pattern. This decoupling was not addressed in former studies where the materials were either permissive to cell adhesion [21,23,25] or not investigated [24]. We thus propose an original protocol for the surface functionalization of polyacrylamide hydrogels that allow to robustly control the uniformity of the adhesive coating in the presence of a rigidity pattern. By guaranteeing a stiffness-independent surface chemistry, we could address the effect of stiffness textures on the tensile stresses that the cells exert to probe the mechanical properties of their surroundings. Adaptation of traction force microscopy to stiffness-patterned substrates then allowed us to measure cellular forces at a submicron resolution and evidence that cells adapt their tensile stresses to micron-scaled stiffness textures.

## 2. Materials and Methods

### 2.1. Fabrication of Stiffness-Patterned Polyacrylamide Hydrogel

Patterned hydrogels were obtained from the rapid photopolymerization of an acrylamide/bis acrylamide solution through a gray-level mask. The preparation of the solution of monomers is described in Reference [33]. In brief, 2 mg of a highly efficient UV-sensitive initiator Irgacure 819 (Ciba Specialty Chemicals, Basel, Switzerland) is dissolved into 10 µL of propylamine (Sigma-Aldrich, Saint Quentin Fallavier, France) at 52 °C for 10 min. A total of 490 µL of deionized water containing 0.22% *v*/*v* of 0.2 µm fluorescent beads (2% solid red beads, Molecular Probes), 250 µL of a 40% solution of acrylamide and 250 µL of a 2% solution of *N*,*N*’-methylene-bis-acrylamide (Bio-Rad, Marnes-la-Coquette, France) are added, leading to a 10–0.5% mixture of monomers.

Hydrophobic gray-leveled chromium masks were engineered. The mask was a copy of the pattern to be transferred to the hydrogel. It was composed of a microscope slide, washed in a Piranha solution of hydrogen peroxide at 30% and concentrated sulfuric acid (Sigma-Aldrich, Saint Quentin Fallavier, France), with a concentration of 1:2 for 10 min. Moreover, 1 nm of titanium and 14 nm of chromium were deposited onto it with a Plassys-type electron gun (MEB550, electron gun 10 kW, Marolles-en-Hurepoix, France). This resulted in a transmission coefficient of 11% at 365 nm. The pattern was etched into the metal deposit using contact optical lithography processes. AZ1512HS resist (MicroChemicals GmbH, Ulm, Germany) was spun onto the metal deposit at 3000 rpm for 30 s to reach a 600 nm thickness. It was then illuminated through a black and white master lithographic mask that reproduced the patterns. The resist pattern was developed with a 1:1 AZ400K-developer (MicroChemicals GmbH, Ulm, Germany):de-ionized water mixture for 1 min. Etching was performed in a DPS type etching reactor [34] (Applied Materials Inc., Santa Clara, CA, USA), using a chlore:oxygen (2:1) plasma (100 sccm/40 sccm 25 mTorr, 600W source, 100W bias). The resist was removed by exposing it 30 s to an oxygen plasma in the DPS reactor. The gray-leveled mask was then rendered hydrophobic by grafting a fluorinated silane onto it [35]: the gray-leveled mask was immersed into a 1‰ solution of Optool DSX (Daikin, Pierre Benite, France) in perfluorohexane (Sigma-Aldrich, Saint Quentin Fallavier, France) for 1 min. It was left to react for 1 h in water vapor at 65 °C, and it was then rinsed for 10 min in perfluorohexane. The gray-leveled chromium mask was then fitted with 40 µm thick wedges on its edges.

The hydrogels were prepared on 30 mm diameter coverslips, chemically activated using Bind Silane (PlusOne, VWR, Rosny-sous-Bois, France) to allow stable, covalent bonding of the polyacrylamide hydrogels, as described in Ref. [33]. A droplet of 30 µL of the precursor solution was deposited onto the activated coverslip and covered with the gray-leveled mask. The hydrogel was reticulated by exposing the precursor solution through the gray-leveled mask to UV-A (2 W/cm^2^, Eleco UVP281, Gennevilliers, France) for a few seconds. The duration of the illumination was varied to tune the mechanical properties of the hydrogel. The hydrogel was then immersed in water and carefully detached from the mask. It was rinsed 3 times in deionized water to remove the unreacted compounds and left for swelling overnight.

### 2.2. Mechanical and Geometrical Characterisation of the Stiffness Patterns

Young’s moduli of the patterned hydrogels were measured on a NanoWizard II AFM (JPK Instruments, Berlin, Germany) in the force mapping mode with MLCT C tip cantilevers (Bruker, Palaiseau, France) with a nominal spring constant of 0.01 N/m. Young’s moduli were fitted from the indentation curves using a Hertz–Sneddon model [36]. Rigidity maps were sampled every 0.3 to 1 µm, depending on the size of the pattern.

The sizes of the rigid dots were determined via comparison to the AFM data. A new image was generated, composed of the sliding average of the AFM data on squares of 3 pixel sides. The limits of the stiff dot were set at places where the gradient of the sliding average was larger than 500 Pa/µm. This region was then filled with the largest disk. The center and the radius of this disk defined the geometric parameters of the stiff dots. The mean of the rigidity was calculated on this area. The average on the 3 to 5 dots was performed and taken as the rigidity of the stiff dots. The limit of the soft region was defined using the same methodology. The largest disk that fills the region limited by the edges of the soft background shares the same center as the disk that fits the stiff dot.

### 2.3. Functionalization of the Stiffness-Patterned Hydrogel

The patterned polyacrylamide hydrogel was made compatible to cell culture by covalent grafting fibronectin to its surface via the photosensitive NHS-ester diazirine heterobifunctional crosslinker sulfo-LC-SDA (sulfosuccinimidyl 6-(4,4′-azipentanamido)hexanoate, Pierce, Thermo Fisher Scientific, Courtaboeuf, France). Fibronectin (Thermo Fisher Scientific, Courtaboeuf, France, reference 33016015) was first crosslinked in PBS (Gibco, Thermo Fisher Scientific, Courtaboeuf, France) to the NHS-ester group of the sulfo-LC-SDA, with a molar ratio of 1/480, following an already described protocol [33]. The compound was then dialyzed against PBS for 48 h to remove the unreacted crosslinker. Moreover, 800 µL of the photosensitive solution of proteins was deposited at different concentrations on the surface of the hydrogel (6 µg/mL to obtain 0.6 µg/cm^2^, or 14.8 µg/mL to obtain 1.7 µg/cm^2^) and incubated for 1 h at room temperature. In order to get a uniform coating, the residual solution was removed and the gel was immediately exposed to UV light for 5 min with a 2 W/cm^2^ UV bulb (Eleco UVP281). Non-uniform coating was obtained by letting the surface of the hydrogel dehydrate for a few minutes to an hour (a longer time ensures that the surface coating is even on the entire patterned region). The gel was then illuminated for 5 min with the 2 W/cm^2^ UV bulb (Eleco UVP281, Gennevilliers, France), and carefully rinsed 3 times with PBS.

### 2.4. Characterization of the Surface Chemistry

The uniformity of the surface density was characterized by immunostaining. The functionalized hydrogel was incubated first with an anti-fibronectin primary polyclonal antibody (Sigma-Aldrich, Saint Quentin Fallavier, F3648, 1 h, room temperature), and then with an Alexa-conjugated secondary antibody A488 (Molecular Probes, Thermo Fisher Scientific, Courtaboeuf, France) (1 h, room temperature). A stack of fluorescence images was acquired with a 40× water immersion objective using an upright Leica SP confocal microscope(Leica Microsystems, Weztlar, Germany). Fourier transform analysis was performed on the thickness-averaged intensity profile using ImageJ software (v 1.53, NIH, USA).

### 2.5. Cell Culture and Immunostaining

Rat embryo fibroblast (REF52) lines stably expressing YFP-paxillin (gift from B. Ladoux, Institute Jacques Monod, Paris, France) were cultured in Dulbecco’s Modified Eagle Medium supplemented with 100 µg/mL glutamine (Gibco, Thermo Fisher Scientific France, reference 31966047) containing 10% fetal bovine serum (Gibco, Thermo Fisher Scientific France, reference 10270106), 100 U/mL penicillin, 100 µg/mL streptomycin (Gibco, Thermo Fisher Scientific France, reference 15240). The cells were maintained at 37 °C in a humidified atmosphere of 5% CO_2_. For the measurements of cellular stresses, cells were cultured in a temperature- and CO_2_-controlled incubation chamber (Okolab, Rovereto, Italy) mounted on a IX83 Olympus-inverted microscope. The experiments started 3 h post-seeding.

### 2.6. Calculations of Cellular Stresses

Cellular stresses were calculated from the deformation field of the fluorescent markers embedded in the hydrogel. Stacks of images with 0.3 µm spacings were acquired with an inverted Olympus IX83 (Olympus France, Rungis, France), equipped with 60× oil immersion objective (NA 1.25) to allow the precise determination of the surface. At the end of the experiment, cells were removed using 0.05% trypsin-EDTA (Gibco, Thermo Fisher Scientific, Gennevilliers, France reference 25300) for 30 min to 1 h to get the reference images of the surface of the gel in the absence of cellular stresses.

#### 2.6.1. Deformation Field

Before calculating the displacement field, images were globally registered for global rotation and translation in x, y, z using autocorrelation of the image of the contractile cell with the reference image (i.e., after trypsin) in four different regions taken as further away as possible from the cell in the corners of the image. From the displacements of these four areas, the rigid registration was calculated and allowed aligning almost perfectly the two images. Bead displacements were calculated using a MATLAB script based on the CR toolbox developed by J. Diener [37,38]. A Kanade–Lucas–Tomasi (KLT) particle tracking algorithm [39] was used to calculate the displacement of each bead. KLT is a pyramidal optical flow method, which was recently shown to be much more accurate and faster than traditional particle image velocimetry techniques (PIV) for TFM [40]. Bead positions are detected using a local maxima search algorithm imposing a minimum distance of 3 pixels in between each points. We were thus able to detect at least 10 beads per stiff dot. Pyramids of images (i.e., smaller resolution images of the initial image and of its spatial gradients) are calculated. The tracking algorithm is successively ran on the different pyramids beginning on the low resolution image, with a large window size. Then the calculated displacement is used back for the next pyramid level to get at each iteration a more accurate quantification of the displacement with a smaller interrogation window around the selected features. We used a pyramid level of 5, and a value of 10 pixels for the last window size.

#### 2.6.2. Traction Force Calculation

The stresses the cells transmit to the matrix were calculated using fast Fourier transform (FFT), following Butler et al. [41]. Before calculating the stresses, displacements were filtered using the *wiener2* function in MATLAB. Sabass *et al.* [42] have shown that such filtering was comparable with using a regularization technique. The FFT method was adapted to the non-uniform Young’s modulus of the hydrogel for the specific case where the Young’s modulus had no in-depth variations (which is the case here, see Appendix A). In the absence of volumetric external stresses, the displacement field $\vec{u}$ in a semi infinite elastic medium results from the derivation of the local mechanical equilibrium:(1)$$\vec{\nabla }\cdot \vec{\nabla }\vec{u}+\left(1-2\nu \right)\Delta \vec{u}=\vec{0}$$

Only the Poisson’s ratio ν of the material comes in Equation (Equation 1), not the Young’s modulus *Y*. Young’s modulus comes in when imposing the surface stresses at the boundaries [43]. Surface stresses can have in-plane spatial variations. Then, as long as the Young’s modulus of the material has no in-depth dependency, changing surface stresses $\vec{f}\left(x,y\right)$ to $\vec{f}\left(x,y\right)/Y\left(x,y\right)$, with (x,y) the in-plane spatial coordinates, does not modify the equations. Following Landau’s derivation, the solution of Equation (Equation 1) on the free surface of the material is:(2)$$\vec{u}=\int G\left(x-x^{\prime },y-y^{\prime }\right)\frac{\vec{f}}{Y}\left(x^{\prime },y^{\prime }\right)\;dx^{\prime }dy^{\prime }$$
with *G* as the Green function:(3)$$G\left(x,y\right)=\frac{1+\nu }{2\pi r}\left(\begin{array}{ccc} 2\left(1-\nu \right)+\frac{2\nu x^{2}}{r^{2}} & \frac{2\nu xy}{r^{2}} & -\frac{\left(1-2\nu \right)}{r}x\\ \frac{2\nu xy}{r^{2}} & 2\left(1-\nu \right)+\frac{2\nu y^{2}}{r^{2}} & -\frac{\left(1-2\nu \right)}{r}y\\ \frac{\left(1-2\nu \right)}{r}x & \frac{\left(1-2\nu \right)}{r}y & 2(1-\nu ) \end{array}\right)$$
where $r=\sqrt{x^{2}+y^{2}}$. Then, the traction stress field $\vec{f}$ writes:(4)$$\vec{f}\left(x,y\right)=Y\left(x,y\right)\cdot F^{-1}\left(F\left(\vec{u}\right)/F\left(G\right))\right)$$
with F denoting the Fourier transformation.

### 2.7. TFM Analysis

Cellular traction stresses are calculated using the deformation field as input, obtained from the analysis of the displacement of the fluorescent markers. Stiff dots and a soft background were analyzed separately, by generating binary masks from the AFM data. The resulting masks were then deformed based on the deformation field to fit with the deformed image (Appendix B). The distribution of the traction stresses was then analyzed on the stiff dots and the soft background. Misalignment of the stress-free reference image as well as image processing for the tracking of the intensity patterns of the fluorescent markers may induce noise in the deformation field. Noise level was quantified by calculating the cellular stresses out of the cell contour. Only cellular stresses that exceeded the 0.95 quantile of the noise were considered significant and accounted for the statistics (see Appendix C).

## 3. Statistical Analysis

Paxillin localization on the stiff dots was obtained from six experiments, with more than 200 cells on a panel of geometries ranging from dot sizes of 2 µm with 2 µm spacing up to 10 µm and a maximal spacing of 15 µm.

TFM was performed on 12 cells (geometries and stiffness reported in Appendix A) from two experiments that were analyzed 3 h after plating, and for three or four time lapses during 1 h 15. As a whole, 39 measurements were exploited. For every pattern condition, the 0.95 confidence intervals around the mean values were calculated by assuming that, for a large number of cells, the distributions of the means were Gaussian. The confidence interval was then obtained from the standard error of the mean and the *p*-value of the bilateral Student *t*-test for the specific degree of freedom:(5)$$ci\left(0.95\right)=t\left(0.05,n-1\right)\frac{s}{\sqrt{n}}$$
with *s* being the standard deviation of the data set and *n* the number of independent cells. Experimental data were fitted using least square weighted fits: mean values of the stresses were weighted by $1/ci{\left(0.95\right)}^{2}$.

## 4. Results

### 4.1. Micron-Scaled Rigidity Patterns with kPa/µm Gradients

In order to address cell stiffness sensing at subcellular scales, we designed culture supports with micron-scaled stiffness patterns based on the formulation of an innovative fast curing, water-based, photosensitive solution of acrylamide/bis acrylamide precursors (Figure 1). A gray-leveled lithography process was set up, illuminating a UV-sensitive solution of acrylamide/bis acrylamide monomers through gray-leveled, patterned photomasks (Figure 1A). Micron-scaled patterns of rigidity could be attained by limiting the diffusion of the entities during the polymerization (Figure 1B). This was achieved by designing an efficient water-optimized photochemistry based on a high absorbance photoinitiator, bis(2,4,6-trimethylbenzoyl)-phenylphosphine oxide (Irgacure 819^®^) that we dispersed in water by the use of propylamine in a vapor phase (see Materials and Methods). We could then print micron-scaled patterns of rigidity with kPa/µm gradients at their edges (Figure 1C). Illumination times were used to tune the reticulation rate of the polyacrylamide. Close-to-linear dependencies of the Young’s moduli of either the stiff dots and the soft background were observed in the range of stiffness of interest (Figure 1D). As the reticulation process occurred in a water-based solution, the diffusion of the monomers and the photoinitiator contributed to the final reticulation [44]. Consistently, we observed that the geometry of the pattern influences the stiffness values. For instance, increased spacing allowed to obtain larger gradients of rigidity between the stiff pattern and the soft background: the stiffness of the stiff dots increased more drastically with longer illumination times while the increase of the stiffness of the background with exposure time was fairly insensitive to the spacing of the stiff dots (Figure 1D and Appendix A).

### 4.2. Tuning of the Uniformity of the Surface Coating by the Hydration Rate of the Hydrogel

As polyacrylamide has an inert surface, the rigidity-patterned hydrogels were coated with the extracellular adhesion protein fibronectin using a very common protocol, consisting of covalently binding the proteins to the surface of the hydrogel through a UV-sensitive crosslinker [6]. However, we observed that the binding of the proteins at the surface of polyacrylamide was very sensitive to the moisture of the surface of the hydrogel. By changing the hydration of the surface of the hydrogel, we could tune the surface density from uniform (Figure 2A) to non-uniform (Figure 2B), with fibronectin condensing on the stiffer pattern. Uniform coating was only achieved by keeping the surface wet. Partial dehydration resulted in an uneven surface density of fibronectin (data not shown) while dehydrating until the surface appeared dry allowed obtaining an even over-functionalization of the stiff dots (Figure 2B). In this regime, the condensation of the fibronectin on the stiff dots was attributed to the migration of the proteins at the surface of the hydrogel driven by capillary forces that pull the surface proteins toward the most hydrated regions. Indeed, stiff polyacrylamide hydrogels dehydrate slower than soft ones (Appendix A); thus, the surface of the stiff dots is expected to retain more water and to remain hydrated for a longer time than the surface of the soft background.

### 4.3. Localization of the Adhesion Proteins on the Stiff Patterns on Uniform Surface Coating

In order to investigate cell response to micron-scaled rigidity patterns, we designed a panel of dot geometries ranging from 2 to 10 µm and stiffness ranging from 0.7 to 20 kPa. The uniformity of the fibronectin coating was achieved by using the “wet” protocol of surface coating, and confirmed using immunostaining and confocal microscopy (Figure 2A). We could thus decouple chemical signals from mechanical ones and prevent non-uniform surface chemistry from activating cell tension reinforcement independent of mechanical cues [45]. We first examined the localization of cell adhesions on these rigidity-patterned matrices. Other studies have already reported that on rigidity-patterned matrices with uniform surface chemistry, adhesion proteins would concentrate on the stiff patterns [24,46]. Reference [24] had also evidenced the existence of a low rigidity threshold, below which the adhesion proteins would no more be influenced by the rigidity pattern. Here, we assayed rat embryo fibroblasts stably expressing yellow fluorescent protein (YFP)-paxillin (REF-52) on the rigidity patterns. In line with the previous studies, we observed that paxillin, a protein of the adhesion plaque [47], condenses in micron-scaled elongated clusters that localize preferentially on the stiff dots (Figure 3 and Video S1). This micron-scaled mechanosensitivity was observed for all of the stiffness conditions, stiff dots ranging from 0.7 to 49 kPa. In addition, the concentration of the adhesion proteins on the stiff patterns was observed on substrates with various surface densities of fibronectin, from 0.6 µg/cm^2^ to 1.7 µg/cm^2^ (Appendix A). Thus, we confirmed that adhesion proteins, such as paxillin or vinculin (data not shown), preferentially localized on the stiff patterns. However, we could not confirm the existence of a low rigidity threshold, a discrepancy that may arise from the different rheological properties of polyacrylamide and Sylgard 184^®^ PDMS, the material used in Reference [24]. From these observations, we conclude that adhesion proteins are sensitive to the local elastic properties of their extracellular matrix, with no evidence of a sensitivity threshold.

### 4.4. Calculation of Cell Traction Stresses on Rigidity-Patterned Substrates

Concentration of adhesion proteins on the stiff micron-sized dots suggests that the adhesion proteins respond to the local rigidity of the extracellular matrix. To reach this organization, the cellular forces that probe the matrix are thus expected to integrate the information on stiffness at the scale of the patterns of rigidity or below. To investigate it, we implemented Traction force microscopy (TFM) on the rigidity patterned substrates. Fluorescent markers were embedded into the hydrogel to probe the deformations of the matrix (see Materials and Methods). Until now, TFM has been implemented for substrates with uniform stiffness. Here, we address rigidity-patterned hydrogels; thus, a non-uniform stiffness profile. We first showed that, in the specific case where the Young’s modulus has no in-depth variations (which is the case here, see Appendix A), the elastic problem of a semi-infinite elastic material submitted to surface stresses has an analytical solution (see Materials and Methods). In brief, for a material with uniform stiffness, the force field that stresses the surface of this material with Young’s modulus *Y* and Poisson’s ratio ν can be straightly calculated in the Fourier space from the measured displacement field $\vec{u}$ [41]. Fourier inversion then allows quantifying the traction stresses $\vec{f}$ from the cell:(6)$$\vec{f}\left(x,y\right)=F^{-1}\left(F\left(\vec{u}\right)/F\left(G\right))\right)$$
where F denotes the Fourier transformation, (x,y) the in-plane coordinates, and *G* the Green function that is the solution of this elastic problem [43]. As detailed in Materials and Methods, we could show that, in the case where Young’s modulus is an in-depth invariant, the elastic problem can still be solved with a similar approach. To this end, the Green function *G* must be normalized by Young’s modulus Y(x,y). The traction stresses are then obtained with the following equation:(7)$$\vec{f}\left(x,y\right)=Y\left(x,y\right)\cdot F^{-1}\left(F\left(\vec{u}\right)/F\left(\tilde{G}\right))\right)$$
with $\tilde{G}=G/Y\left(x,y\right)$, *G* being the same Green function as in Equation (Equation 6).

### 4.5. Cells Pull on the Rigidity Patterns Proportional to the Local Stiffness

Our aim was to probe cell traction stresses on rigidity patterns with micron sizes. In order to resolve these micron-scaled modulations of the rigidity, we developed an algorithm that could detect the fluorescent markers embedded in the hydrogel with a submicron resolution. An optical flow algorithm based on the Kanade–Lucas–Tomasi feature tracker was implemented to measure the displacement of the fluorescent markers at the surface of the hydrogel (see Materials and Methods). This original algorithm allowed us to reach a resolution of 600 nm, permitting the measurement of the displacement field of at least 10 points on every stiff dot. Using this highly resolved detection and a TFM algorithm adapted to patterned stiffness (Equation (Equation 7)), we could thus quantify the traction stresses, $\vec{f}$. We observed significant stresses both on the stiff dots and on the soft background (Figure 4C). Larger stresses were measured on the stiff dots that support paxillin-stained adhesions (Figure 4D), but the stress amplitude on the soft part of the matrix significantly exceeded the background noise (Figure 4E and Figure A2). This observation suggests that the cells pull with larger stresses on the stiff regions than on the soft ones, although they also probe the soft ones with local tractions.

Measurements of traction stresses were performed on a panel of 5 distinct conditions of stiffness and geometry, with entangled couples of Young’s moduli for the soft background and stiff dots (Appendix A). By averaging the traction stresses of all cells on the stiff and the soft regions for every rigidity pattern, we observed that the traction stresses correlate linearly with the local rigidity ($r^{2}$ = 0.96) (Figure 4F). This observation indeed supports the idea that cell adhesions possess an intrinsic, adhesion-scaled, contractile machinery, as already suggested by others [48,49,50]. It additionally shows that this machinery probes the extracellular matrix with stresses that linearly adapt to micron-sized rigidity modulations, at least in the range we explored, from 0.4 to 12.4 kPa.

## 5. Discussion

By developing a technology of stiffness patterning in soft hydrogels, together with a decoupled surface chemistry, we could address single cell rigidity sensing at the micron scale.

Polyacrylamide hydrogels with kPa/µm gradients were designed (Figure 1C), which showed sharp stiffness textures with differences of several kPa between the stiff, micron-scaled dots and the soft background. This performance was previously attained with photodegradation methods [46] or e-beam reticulation [24]. However, it was never attained with the widely used technique of UV photo-polymerization. More specifically, 1-[4-(2-hydroxyethoxy)-phenyl]-2-hydroxy-2-methyl-1-propane-1-one (Irgacure 2959^®^ Ciba Specialty Chemicals, Basel, Switzerland), a photo-initiator that is commonly used in polyacrylamide photochemistry [51,52,53], and that, in principle, suits to water-based acrylate resists, did not allow printing gradients sharper than 0.7 kPa/µm [54]. This new photochemistry indeed allowed to reduce exposure times to few seconds, compared to several tens of seconds that would be required with Irgacure 2959^®^. This results in the decrease of the diffusion during photopolymerization, which blurs the stiffness gradients. We could thus reproduce rigidity profiles with gradient amplitudes that resemble those measured on human tissues [19,20,55,56].

A new polyacrylamide surface functionalization protocol has also been proposed, which allows controlling the distribution of the extracellular matrix proteins on the stiffness pattern. In the context of measuring cell mechanosensitivity to micron-scaled stiffness texture, a stiffness-independent surface density was achieved by maintaining the surface of the hydrogel wet (Figure 2A). However, a non-uniform coating could also be obtained by dehydrating the surface of the hydrogel, fibronectin then having a larger surface density on the stiff dots (Figure 2B). Dehydration was already proposed to pattern nanoparticles on hard materials by using topography-controlled dewetting in between the substrate and a mold [57]. Here, the concentration of the molecules on the stiff patterns also originates from the capillary forces involved in dewetting, but the dewetting is tuned by the stiffness-dependent kinetics of evaporation at the surface of the hydrogel (Appendix A). This original protocol based on the dehydration of the hydrogel, while performing the coating, opens new perspectives to pattern molecules at the surface of hydrogels. Compared to stamping methods [58] or local surface activation [59,60], its handling is much simpler. The limitation is however that the surface density of the proteins is not zero outside the pattern, as expected with the other techniques. Thus, cell confinement may be less robust for long cultures, although long-term cell maintenance on patterns is a difficult problem that is also shared by the other techniques. Here, unlike the other techniques, the surface density of the coating is coupled to the local stiffness of the hydrogel. This coupling makes sense when the goal is to engineer biomimetic extracellular matrices, as in vivo variations in the density of the extracellular matrix are observed and result in gradients of rigidity [17].

The implementation of a rigidity-independent surface chemistry made it possible to investigate the influence of periodic stiffness patterns on cell mechanical interaction with its matrix. Assaying REF 52 fibroblast cells stably transfected with YFP paxillin, we observed that the adhesion protein paxillin (and also vinculin, data not shown) condenses preferentially on the stiff patterns, whatever the rigidity difference with the soft background or the geometric details of the pattern (Figure 3, Appendix A). Preferential localization of integrin-dependent adhesions on stiff, micron-scaled patterns had already been reported in mesenchymal stem cells [24,46]. Here, different from a previous work [24], we could not report on a sensitivity threshold below which cell adhesions loose sensitivity to rigidity. This discrepancy was attributed to differences in the rheological properties of the materials at low elastic modulus. Contrarily to polyacrylamide, PDMS used in Reference [24] is known to become viscous when lowering the rigidity, enabling cells to remodel the extracellular matrix [61].

Traction stresses were then measured, to evaluate how cells interact mechanically with the subcellular modulations of the stiffness of the extracellular matrix. This was achieved by generalizing TFM to substrates with a non-uniform rigidity. Additionally the detection of the displacement field at the surface of the substrate was improved by implementing an optical flow algorithm to track the fluorescent markers. This algorithm made it possible to resolve the displacement field inside the micron-scaled stiffness patterns, with a performance similar to that obtained with the more advanced, but more demanding, technique of stimulated emission depletion microscopy (STED) [62]. More than 10 beads could be tracked in every stiff dot, allowing a precise determination of the traction stress field.

As a result, we found that the cells actually probe the local, micron-scaled, mechanical properties of the matrix with traction stresses that are proportional to the local rigidity, at least in the range of rigidity we explored, from 0.4 to 12.4 kPa (Figure 4F). This observation was consistent with data reported at the cell scale, on substrates with uniform stiffness [63,64]. Larger stresses were thus observed on the stiff dots, more specifically in places of large paxillin-stained focal adhesions (Figure 4D). Significant traction stresses were also measured on the soft background, out of paxillin-stained adhesions (Figure 4D–E). This observation questions methodologies of traction force calculation, which constrain cellular forces to colocalize with vinculin or paxillin-stained focal adhesions [65]. However, the observation of significant stresses out of paxillin-stained adhesions is indeed fully consistent with Reference [66], which reports that cell-matrix contacts may be negative to paxillin or vinculin staining and involve other molecules as a connector to actin, for instance tensin. In addition, it should be noted that the patterns of rigidity that were used here did not induce noticeable change in cell shape. In these experiments, cell geometry was then not expected to significantly influence the amplitude of the traction stresses as already reported [67,68].

Our observations thus lead to consider that cells probe the stiffness of the extracellular matrix down to the micron scale. Several studies have already pointed the existence of an adhesion-scaled mechanism to probe the stiffness of the matrix, either by reporting on molecular complexes involved in rigidity sensing [48,49,69,70] or by providing predictions based on theoretical modelling [71,72,73,74]. Cell-scaled stiffness sensing has also been proposed [75,76,77]. Although not addressed here, some of the results that were obtained raise comments in regard to our work. References [76,77] addressed this using elastomeric posts. The elastomeric posts are made of PDMS and have a local stiffness of the order of MPa, the bulk—Young’s modulus of PDMS. Should our observation be consistent in these devices, the measured traction stresses should be constant, independent of the bending rigidity, as the local rigidity is given by the bulk elastomer. This is definitely not what experiments on elastomeric posts report [76,78,79]. In these device, the cell’s mechanical response is clearly tuned by the bending rigidity of the posts. One reason could be that, in the case of the elastomeric posts, adhesions are more mature than on the soft hydrogels because of the larger local stiffness of the bulk elastomer. Actin also mostly organizes as stress fibers [77], while this is not the case on hydrogels [80]. The mechanism of rigidity sensing that is investigated on elastomeric posts may therefore correspond to a specific maturity of cell adhesion, with mature focal adhesions and actin cytoskeleton organized as stress fibers. The adaptation of actin’s structuring to the stiffness at the cell scale on continuous supports [80] and its fixed structure in stress fibers on elastomeric posts [77] could be at the origin of this very different behavior.

## 6. Patents

This work has resulted in two patents: WO2013079231 and WO2019166487.

## Figures and Tables

**Figure 1 nanomaterials-12-00648-f001:**
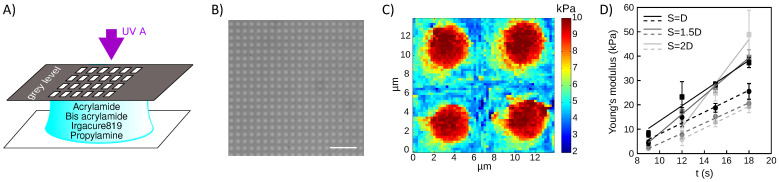
Gray-level photolithography of enhanced UV-sensitive solution of acrylamide/bis acrylamide monomers allowed to print micrometer-scaled patterns of rigidity with kPa/µm gradients. (**A**) The technological process used an amine additive to dissolve the photoinitiator in water. (**B**) DIC image of 3.5 µm dots with spacing 4 µm (pitch 7.5 µm): the rigidity modulation resulted in an optical index modulation. Bar 30 µm. (**C**) Rigidity map of (B) obtained with AFM indentation measurements: rigid dots are 9.1 ± 1.2 kPa, soft background is 5.1 ± 0.9 kPa; gradients at the edges of the dots are around 5 kPa/µm. (**D**) Sensitivity of the Young’s moduli of the dots (—) and of the background (- - -) to the illumination time, in dependence on the geometry of the pattern. Dots are D = 5 µm wide, and are spaced by S = 5, 7.5 or 10 µm.

**Figure 2 nanomaterials-12-00648-f002:**
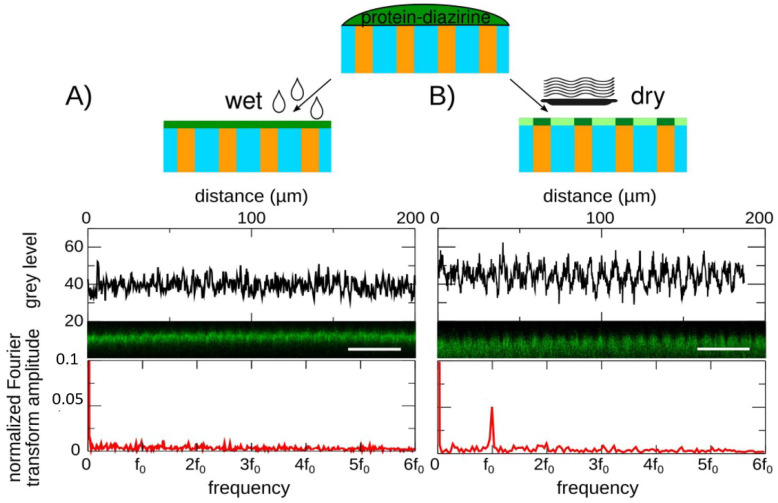
Tuning the dehydration of the surface of the hydrogel controls the distribution of the surface coating. (**A**) Maintenance of the hydration of the surface allows a uniform surface coating, as shown by the intensity analysis of labeled fibronectin coated on the hydrogel: confocal cross-section, and Fourier transform reveal a flat intensity profile. (**B**) Surface dehydration results in the over-functionalization of the stiff regions of the hydrogel (dehydration time: 45 min). Confocal cross-section and Fourier transform of the intensity profile reveal periodic brighter spots ($f_{0}=1/7.5$ μm^−1^). Blue and orange in the schematics stand for the soft and stiff parts of the stiffness pattern. Bars: 30 µm.

**Figure 3 nanomaterials-12-00648-f003:**
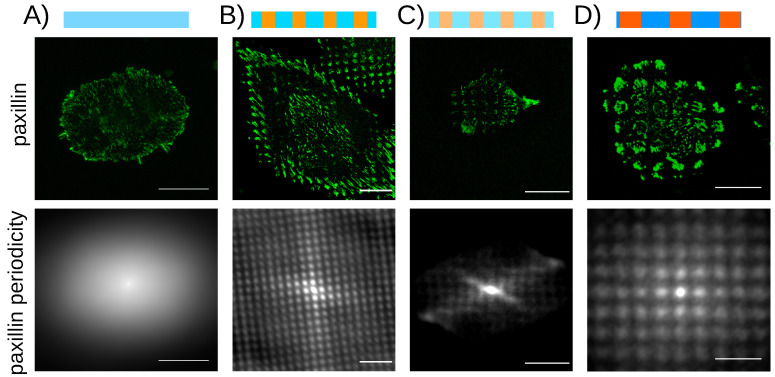
The adhesion protein paxillin condenses on the rigid dotted pattern, whatever the explored rigidities or geometries of the pattern. (**A**–**D**) Top sketches illustrate the geometry and the stiffness of the patterns. Blue and orange represent the soft and stiff parts. Darker colors stand for “larger” stiffness. (**A**) Control behavior on a uniformly soft hydrogel of 3.4 kPa. The autocorrelation function of the intensity shows no periodicity. (**B**–**D**) Top: Paxillin distribution. Bottom: the autocorrelation function of the intensity shows the periodic pattern of the substrate. Geometric and elastic parameters of the stiff dots and soft spacings: (**B**) (3.5 µm, 9.1 kPa)–(4 µm, 5.1 kPa); (**C**) (3.5 µm, 0.7 kPa)–(4 µm, 0.4 kPa); (**D**) (5 µm, 14.6 kPa)–(7.5 µm, 6.5 kPa). Bars: 30 µm.

**Figure 4 nanomaterials-12-00648-f004:**
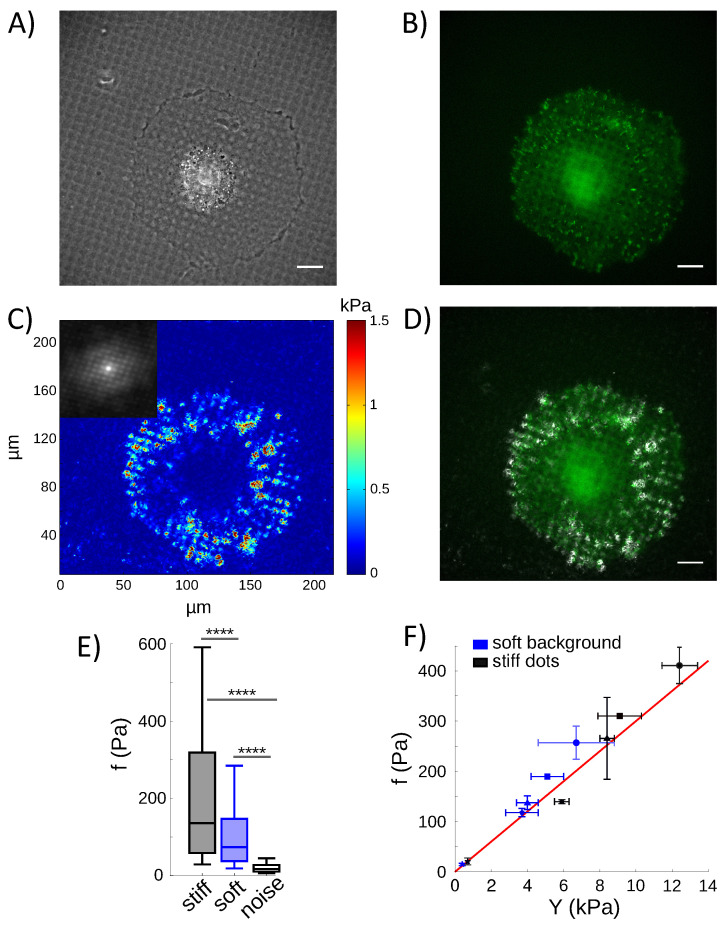
(**A**) Phase contrast image of a REF52 cell on the rigidity-patterned matrix shown in Figure 1. (**B**) Paxillin concentrates mainly on the dots. (**C**) Amplitude of the traction stresses. The amplitude of the stresses distributes periodically (see the autocorrelation function in the inset). (**D**) Large stress amplitude (in white) colocalizes with high intensity paxillin staining (in green). Bars: 30 µm. (**E**) Stress amplitude for this single cell. Stresses are significantly larger on the stiff dots than on the soft background, but the latter also significantly emerges from the noise (Student paired sample *t*-test, $p<10^{-10}$, denoted by ****). (**F**) Cells-averaged traction stresses linearly correlate with the local Young’s modulus of the hydrogel (red line, $r^{2}=0.96$, slope $\left(30.0\pm 4.2\right)\times 10^{-3}$). Black (resp. blue) symbols: stiff dots (resp. soft background). Pairs of rigidity are represented with the same symbol (•, ▪, ▴, ⧫, ★). Bars are the standard error of the mean.

## Data Availability

Data presented in this article is available on request from the corresponding author.

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
