# Peer review of "Cells on Hydrogels with Micron-Scaled Stiffness Patterns Demonstrate Local Stiffness Sensing"

_nanomaterials, 2022, doi:10.3390/nano12040648_

Round 1

Reviewer 1 Report

Authors reported the investigation of whether cells are able to sense micron-scale stiffness texture by measuring the forces they transmit to the extracellular matrix. The performance sounds good and a series of results were also discussed. However, the innovation in this paper is not very well put forward, some issues should be addressed.

1, the writing in Introduction part is not good. I recommend rewriting and exhibit your innovation.

2, Please remove summary of the results from the final paragraph of the introduction section.

3, The mechanical property measurement should refer to the standard. I suggest authors add the related standard or reference to standardize the test.

4, the stiff pattern on uniform surface coating is a determinant in applications of micro- and nanoarrays. How to improve the controllable patterns and arrays in this work? The authors should also pay attention to this challenge, and some pioneering and original researches about controllable assembly of nanoparticles are suggested: Giant, 2021, 8, 100076; Adv. Mater. 2017, 29, ‪1703143;

5, By using the fabrication method in this article, how small can the minimum size of microstructures assembled by proteins? How to improve the fabrication resolution in this method?

6, At least, some applications based on protein assembly structures should be proposed. And this work would be further enhanced if any applications could be demonstrated.

Reviewer 2 Report

The paper deals with interesting issues of cell biology and the topic itself is worthy of publication unfortunately the descriptions of the implementation leave much to be desired.

The authors in the abstract declare" Additionnaly, an original protocol for the surface coating of adhesion proteins is proposed" however, the description of the syntheses presented in this work is vague. If it is not clarified the publication cannot be published. Additionally, the introduction is very laconic. The authors could write more about the techniques they use and the practical application of their observations. Below are some more detailed comments

The materials and methods section lacks information about the purity of the reagents. Information about the company where the purchase was made is also incomplete, missing country and city.

Line 79 „Piranha solution with concentration 1:2” I guess the author meant the volume ratio of perhydrol to sulfuric acid, but this should be written directly.

Line 80 „Plassys type electron gun” please include precise information about the equipment.

Line 86” in a DPS type etching reactor” please describe your equipment precisely

Line 98 “for few seconds” exactly how much

The whole process of preparing the material is complex, it would be useful to have some scheme of its obtaining and functionalization

Line 129 „few minutes to an hour” how long a few minutes or an hour?

Line 154 „removed using 1x trypsin (Lonza)” What was the trypsin concentration or with EDTA such laconic information is unacceptable

Round 2

Reviewer 1 Report

All issues were addressed. 

Reviewer 2 Report

Thank you for engaging in discussion with the reviewer.